# Supply chain product quality control strategy in three types of distribution channels

Lilong Zhu [1,2]*

**1** School of Business, Shandong Normal University, Ji'nan, Shandong, China, **2** School of Management, Shandong University, Ji'nan, Shandong, China

* zhulilong2008@126.com

**Data Availability Statement:** All relevant data are within the manuscript. No supporting information files.

**Funding:** This work was supported by the Humanities and Social Sciences Foundation of the Ministry of Education in China under grant No.

## Abstract

Based on a three-stage stackelberg dynamic game analysis, this paper constructs a product quality control strategy model for three types of distribution channels (direct channel, retail channel and mixed channel) in a three-echelon supply chain, which is composed of one manufacturer, one retailer and the final customer. This paper studies how to design a distribution channel strategy and provides a product quality control strategy. Furthermore, this paper analyzes three types of distribution channels strategy in the context of how they influence a manufacturer's product quality decision and quality prevention strategy, a retailer's product pricing decision and quality inspection strategy, and the final customer's product demand decision. We compare the manufacturer's product quality level, quality prevention effort level, wholesale price, direct sale price and the retailer's quality inspection effort level, retail price in three types of distribution channels and determine the manufacturer's and retailer's expected profits function and the final customer's consumer surplus. In addition, we introduce the distribution channels demand elasticity ratio to analyze the influence of determining the product quality control strategy. Most importantly, we conduct a numerical sample analysis that will prove the model's effectiveness and indicate a specific application in practice.

## 1 Introduction

In recent years, with the rising of network economy and e-commerce, in addition to the traditional retail channel, more and more customers or consumers choose to purchase products from the internet channel(direct channel), which have become an important way of products sale. With the changing in customer or consumer buying behavior, more and more companies are beginning to redesign or rebuild their distribution channel structure(Chiang W et al. 2003 [1], Tsay A et al. 2004 [2], Kenji M 2017 [3], Yan W et al. 2018 [4]), Such as HP, Nike, Lenovo, in addition to focus on the traditional retail channel, have also opened up an internet channel (direct channel); Dell, MI has been focused on internet channel in the past, and now also began selling products in traditional retail channel; and Apple, Haier sell their products in the traditional retail channel and internet channel in the same time, which used a mixed channel structure. Many facts have proved that the mixed channel structure which composed of the

17YJA630147, Nature Science Foundation of Shandong Province under grant No. ZR2019MG017 and National Social Science Foundation of China under grant No. 13AGL012 to LZ.

**Competing interests:** The authors have declared that no competing interests exist.

traditional retail channel and the internet channel(direct channel), on the one hand can achieve better customer coverage and penetration(Jerath K et al. 2017 [5], Tian L et al. 2018 [6]), on the other hand may also lead to different distribution channels conflict, competition and imbalance(Chen J et al. 2017 [7], Lan Y et al. 2018 [8]).

Nowadays, more and more researchers focus on how to design a distribution channels strategy and determine a product quality control strategy in different types of distribution channels in a three echelon supply chain that is composed by one manufacturer, one retailer and the final customer, which have become one of hot research fields in supply chain management. However, nowadays the research field has three potential systemic problems: first of all, how to design different types of distribution channels structure in a three echelon supply chain (direct channel, retail channel and mixed channel); what's more, the different types of distribution channels structure in supply chain how to influence the manufacturer's product quality decision and quality prevention strategy, the retailer's product pricing decision and quality inspection strategy, and the final customer's product demand decision; above all, how to influence the manufacturer's and retailer's expected profits function and the final customer's consumer surplus, and how to determine a product quality control strategy in order to eliminate "channel conflict" and "free-riding behavior". All of these problems and difficulties have not been fully resolved, which are also important research directions for current researchers.

In this paper, we will construct a distribution channel strategy model in a three echelon supply chain that is composed of one manufacturer, one retailer and a final customer based on a three-stage stackelberg dynamic game. Furthermore, we will introduce the distribution channel demand elasticity ratio and investigate how to craft a product quality control strategy in three types of distribution channels (direct channel, retail channel, and mixed channel), which will eliminate the influence of "channel conflict" and "free-riding behavior". Most important, we will determine the manufacturer's product quality level, quality prevention effort level, wholesale price, direct sale price, and the retailer's product quality inspection effort level and retail price, the manufacturer's and retailer's expected profits function, and the final customer's consumer surplus. Then, we will conduct a numerical sample analysis that will indicate a specific application in practice.

## 2 Related literature

This paper is chiefly related to three streams of literature. The first stream is the research on how to design a distribution channels structure strategy, the different types of distribution channels structure and how to influence the product quality decision in a supply chain. Yunchuan Liu (2011) [9] established a channel model to analyze the benefits of competitive upward channel decentralization. Anastasios X (2012) [10] studied how to apply optimal newsvendor policies for a dual-sourcing channel in a supply chain. Hongyan Shi et al. (2013) [11] analyzed consumer heterogeneity and product quality and how to influence the coordination of distribution channels. Guangye Xu et al. (2014) [12] constructed a two-way revenue contract to coordinate a dual-channel supply chain. Salma Karray (2015) [13] investigated how vertical strategy and horizontal strategy influence cooperative promotions in the distribution channel. Kenji M (2016) [14] investigated the optimal product distribution strategy for a manufacturer that used dual-channel supply chains. Kinshuk J et al. (2017) [15] discussed how to make a product quality level decision in a distribution channel with demand uncertainty. Liu Yan et al. (2018) [16] provide insights on how market size uncertainty affects the optimal quality and quantity provision in distribution channels. Ranjan A and Jha J (2019) [17] investigate the pricing strategies, green quality and coordination mechanism between the members in a dual-channel supply chain.

The second stream pertains to designing a product quality contract and establishing a quality incentive mechanism in a supply chain. Peng Ma et al. (2013) [18] created a product quality contract design for two-stage supply chain coordination through integrating manufacturer-quality and retailer-marketing efforts. Jie Zhang et al. (2014) [19] discussed a strategic pricing method with reference effects in a quality competitive supply chain. Raaid B et al. (2016) [20] analyzed the effect of adopting a dual-channel on the performance of a two-level supply chain. Chen J et al. (2017) [7] consider the supply chains can be centralized or decentralized, and demonstrate that quality improvement can be realized when a new channel is introduced. Li Wei and Chen Jing (2018) [21] develop game-theoretic models in which the retailer sells a product in two quality-differentiated brands to demonstrate that the quality difference. Zhang J et al. (2019) [22] use an analytical model to study the interrelationship between a platform's contract choice and a manufacturer's product quality decision.

The third stream of related literature concerns research on product quality risk sharing and the quality strategy of distribution channels in a supply chain. Zhu Lilong et al. (2011) [23] explored manufacturers' moral hazard strategy and quality contract design in a two-echelon supply chain. Cinzia B et al. (2012) [24] discussed product quality-driven innovation with the design of a quality control contract. Christina Wong et al. (2013) [25] investigated the combined effects of internal and external supply chain integration on product quality innovation. Rui H and Lai G (2015) [26] investigated the deferred payment and inspection mechanisms for mitigating supplier's product quality risk. Xiao T and Jim Shi (2016) [27] studied a manufacturer marketing a product and considered the pricing and channel priority strategies of dual-channel supply chain. Wang S.J et al. (2017) [28] explore interaction of channel structure with price-and quality-based competition between two manufacturers. Lin T and Jiang B (2018) [29] discussed the effects of consumer-to-consumer product sharing risk and profit on different distribution channel structure.

In this paper, first of all, we will introduce the distribution channel demand elasticity ratio and investigating how to construct a product quality control strategy model and channel coordination in three types of distribution channels (direct channel, retail channel, and mixed channel); what's more, we consider the manufacturer's product quality decision and quality prevention strategy, the retailer's product pricing decision and quality inspection strategy, and the final customer's product demand decision in a three-echelon supply chain; above all, we also establish a product quality control strategy model in three types of distribution channels to eliminate the influence of "channel conflict" and "free-riding behavior", which will improve the manufacturer's and retailer's expected profits and the final customer's consumer surplus.

The remainder of our paper is organized as follows: in section 3, we describe the model and the basic assumption; in section 4, we consider the product quality strategy in the direct channel and determine the first-best contact parameters; in section 5, we investigate the product quality strategy in the retail channel and establish the manufacturer's and retailer's stackelberg "leader-follower" quality control model, and we compare the contract parameter differences with the direct channel. In section 6, we investigate the product quality strategy in the mixed channel that includes a retail channel and a direct channel scenario simultaneously, and in section 7 we present a numerical example analysis to verify our model results. Finally, we provide the research conclusions and direction for future research.

## 3 The model and assumption

In this paper, we establish a three-echelon supply chain structure that consists of a risk-neutral manufacturer and retailer, and the final customer. The manufacturer first makes decision of the product quality. $q_i$ is the manufacturer's product quality level; furthermore, $i \in \{D,R,MC\}$

denote the direct channel, the retail channel and the mixed channel respectively. The product quality cost function is $C_i(q_i) = kq_i^2/2$ ($k$ is the manufacturer's production technology elasticity); so, we assume $C_i'(q_i) > 0$, $C_i''(q_i) > 0$ and $C_i(0) = C_i'(0) = 0$, $C_i'(+\infty) = +\infty$, i.e. $C_i(q_i)$ is the convex function of increasing marginal cost. $\lambda_m$ is the manufacturer's product quality prevention effort level, and $\lambda_m \in [0,+\infty)$; then, we can obtain that the manufacturer's product quality prevention level is $(1 - e^{-\lambda_m})$. Furthermore, $(1 - e^{-\lambda_m}) \in [0, 1]$, and the corresponding manufacturer's quality prevention cost function is $(1 - e^{-\lambda_r})C_m(\lambda_m) = \eta_m \lambda_m$, $\eta_m$ is the manufacturer' quality prevention cost elasticity. The parameter $w$ is the manufacturer's wholesale price, $P_D$ is the direct sale price in a direct channel, and $T$ is the manufacturer's transfer payment to the retailer in order to eliminate the manufacturer's and retailer's channel conflict.

The retailer purchases the product from the upstream manufacturer and makes decision of the product quality inspection. $\lambda_r$ is the retailer's quality inspection effort level, and $\lambda_r \in [0, +\infty)$; then, the retailer's product quality inspection level is $(1 - e^{-\lambda_r})$. Furthermore, $(1 - e^{-\lambda_r}) \in [0, 1]$, and the corresponding retailer's quality inspection cost function is $(1 - e^{-\lambda_r}) C_r(\lambda_r) = \eta_r \lambda_r$, $\eta_r$ is the retailer's quality inspection cost elasticity. The parameter $p_R$ is the retailer's retail price.

The final customer's quality utility is $\theta q_i$, and $\theta$ denotes the type of final customer; then, we assume $\theta \sim U[a, b]$ uniform distribution, i.e. $a$ is the final customer lower limit of distribution quantity, $b$ is the final customer upper limit of distribution quantity, and the corresponding final customer's consumer surplus is $(\theta q_i - p_i)$.

The final customer's product demand function will be $D_i(q_i) = \alpha - \beta_i p_i/q_i$; $\alpha$ denotes the market maximum demand, and $\beta_i$ is the product demand price elasticity.

In this paper, the manufacturer will determine the three types of distribution channels including a direct channel, a retail channel and a mixed channel. The three-stage stackelberg game is in the following order: in stage one, the manufacturer determines the product quality level in a different distribution channel and determines the product quality prevention effort level; in stage two, the manufacturer determines the wholesale price in a retail channel or the direct sale price in a direct channel; and in stage three, the retailer determines the product quality inspection effort level and the retail price.

And then, the three types of distribution channels decision system is described as Fig 1.

## 4 Product quality strategy in direct channel

In the direct channel, the manufacturer sells its product to the final customer directly through an internet or online ordering system; then, the manufacturer determines the product quality level, the quality prevention effort level and the direct sale price. Therefore, the manufacturer's expected profits' function model is as follows.

$$Max \Pi_M^D(q_D, \lambda_m, p_D) = (p_D - kq_D^2 / 2)(\alpha - \beta_D p_D / q_D)(1 - e^{-\lambda_m}) - \eta_m \lambda_m \qquad (1)$$

The manufacturer's decision variables are the product quality level $q_D$, the quality prevention effort level $\lambda_m$ and the direct sale price $p_D$.

**Proposition 1** In the direct channel, with the final customer's product demand price elasticity decreases, the manufacturer's product quality level and direct sale price will increase, and the quality prevention effort level will also increase. In this scenario, the manufacturer's expected profits' function is concave; i.e. an optimal product quality level exists that will to be maximum. Then, the final customer's consumer surplus will increase with the decreasing in the demand price elasticity.

**Proof.** Based on the stackelberg game analysis, this paper will use the backward induction method to solve the equation. Thus, using the first-order and second-order optimal condition

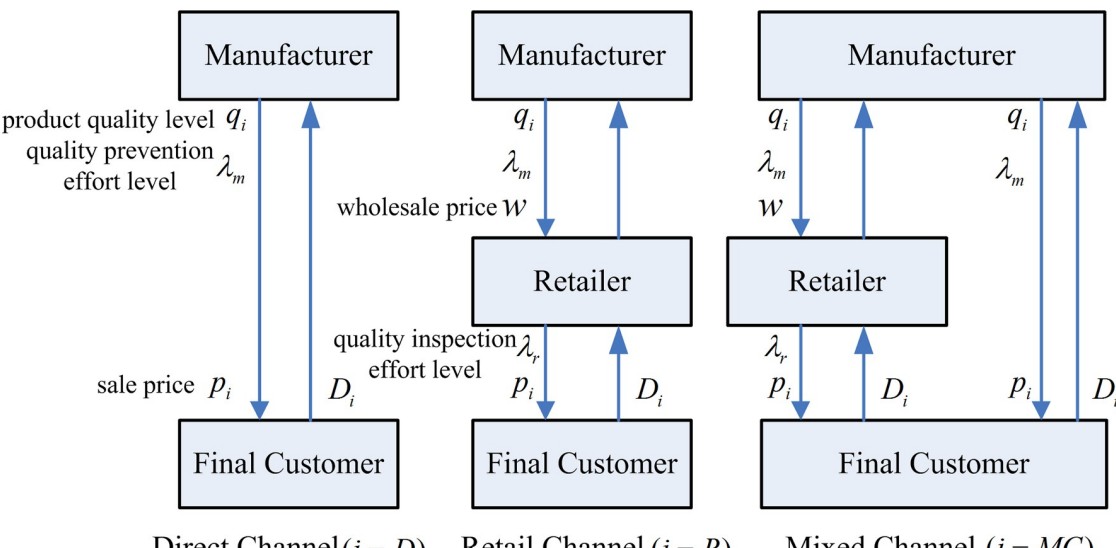

**Fig 1. Three types of distribution channels decision system.**

with respect to $p_D$ in formula (1) yields the following:

$$\partial \Pi_M^D / \partial p_D = (\alpha - \beta_D p_D / q_D)(1 - e^{-\lambda_m}) + (p_D - k q_D^2 / 2)(-\beta_D / q_D)(1 - e^{-\lambda_m}) = 0 \quad (2)$$

$$\partial^2 \Pi_M^D / \partial p_D^2 = (-\beta_D / q_D)(1 - e^{-\lambda_m}) + (-\beta_D / q_D)(1 - e^{-\lambda_m}) < 0 \quad (3)$$

$$\therefore p_D = (\alpha q_D + k \beta_D q_D^2 / 2) / 2 \beta_D \quad (4)$$

Then, we substitute Eq (4) into formula (1) and use first-order and second-order optimal conditions with respect to $\lambda_m$, which yields the following:

$$Max \Pi_M^D(q_D, \lambda_m) = (\alpha q_D / 2 \beta_D - k q_D^2 / 4)(\alpha / 2 - k \beta_D q_D / 4)(1 - e^{-\lambda_m}) - \eta_m \lambda_m \quad (5)$$

$$\partial \Pi_M^D(q_D, \lambda_m) / \partial \lambda_m = (\alpha q_D / 2 \beta_D - k q_D^2 / 4)(\alpha / 2 - k \beta_D q_D / 4) e^{-\lambda_m} - \eta_m = 0 \quad (6)$$

$$\partial^2 \Pi_M^D(q_D, \lambda_m) / \partial \lambda_m^2 = -(\alpha q_D / 2 \beta_D - k q_D^2 / 4)(\alpha / 2 - k \beta_D q_D / 4) e^{-\lambda_m}$$
$$< 0 \text{ (concave function)}$$

Therefore, we derive that

$$\lambda_m{}^D = \ln (\alpha q_D / 2 \beta_D - k q_D^2 / 4)(\alpha / 2 - k \beta_D q_D / 4) / \eta_m \quad (7)$$

Based on the above analysis, we conclude that $p_D$ and $\lambda_m{}^D$ is the manufacturer's first-best sales price, and the quality effort level occurs with a direct channel.

Thereafter, we use the first-order and second-order optimal conditions with respect to $q_D$ in Eq (5), which yields the following:

$$\partial \Pi_M^D(q_D) / \partial q_D = (\alpha / 2\beta_D - kq_D / 2)(\alpha / 2 - k\beta_D q_D / 4)(1 - e^{-\lambda_m})$$

$$-(\alpha q_D / 2\beta_D - kq_D^2 / 4)(-k\beta_D / 4)(1 - e^{-\lambda_m}) = 0 \tag{8}$$

$$q_D = 2\alpha / 3k\beta_D \text{ or } q_D = 2\alpha / k\beta_D \tag{9}$$

$$\because \partial^2 \Pi_M^D(q_D) / \partial q_D^2 < 0 \text{ (concave function)}$$

$$\therefore q_D < 4\alpha / 3k\beta_D \tag{10}$$

Combine Eqs (9) and (10), we derive that

$$q_D^* = 2\alpha / 3k\beta_D \tag{11}$$

$$\partial q_D^* / \partial \beta_D = -2\alpha / 3k\beta_D^2 < 0 \text{ (monotonically decreasing function)}$$

We substitute Eq (11) into Eqs (4) and (7), respectively, to obtain the following

$$p_D^* = 4\alpha^2 / 9k\beta_D^2, \ \lambda_m^{D*} = ln2\alpha^3 / 27k\eta_m\beta_D^2 \tag{12}$$

$$\partial p_D^* / \partial \beta_D = -8\alpha^2 / 9k\beta_D^3 < 0, \ \ \partial \lambda_m^{D*} / \partial \beta_D = -2 / \beta_D < 0$$

Thereafter, we substitute Eqs (11) and (12) into formula (1), and we obtain that

$$\prod_M^{D*} = 2\alpha^3 / 27k\beta_D^2 - \eta_m(1 + \ln 2\alpha^3 / 27k\eta_m\beta_D^2) \tag{13}$$

Therefore, we can describe the final customer's consumer surplus as follows:

$$\therefore CS^{D*} = \int_a^b (\theta q_D^* - p_D^*)f(\theta)\,d\theta = (a + b)\alpha / 3k\beta_D - 4\alpha^2 / 9k\beta_D^2 \tag{14}$$

$$\therefore \partial CS^{D*} / \partial \beta_D = -(3(a + b)\beta_D\alpha - 8\alpha^2) / 9k\beta_D^3 < 0 \text{(monotonically decreasing function)}$$

## QED

Based on proposition 1, we conclude that, in the direct channel, the manufacturer's product quality level, the direct sale price and the quality prevention effort level will increase with a decrease in the final customer's product demand price elasticity. In addition, the manufacturer's expected profits function is concave, and $q_D^*$ and $\prod_M^{D*}(q_D^*)$ is the manufacturer's optimal quality level and maximum expected profits. Thereafter, the final customer's consumer surplus will increase with the decrease in demand price elasticity.

## 5 Product quality strategy in retail channel

In the retail channel, the manufacturer sells product to the retailer, which will determine a product quality inspection strategy and then sell the product to the final customer. The manufacturer determines the product quality level, the quality prevention effort level and the wholesale price, and the retailer determines the quality inspection level and the retail price. Therefore, the manufacturer's and retailer's stackelberg "leader-follower" control model is

described as follows:

$$MaxΠ_M^R(q_R, λ_m, w) = (w - kq_R^2 / 2)(α - β_R p_R / q_R)(1 - e^{-λ_m}) - η_m λ_m \quad (15)$$

$$s.t.\{λ_r, p_R\} = \arg MaxΠ_R^R(λ_r, p_R)$$

$$MaxΠ_R^R(λ_r, p_R) = (p_R - w)(α - β_R p_R / q_R)(1 - e^{-λ_r}) - η_r λ_r \quad (16)$$

Therefore, formula (15) is the manufacturer's expected profits function; formula (16) is the retailer's expected profits function.

**Proposition 2** In the retail channel, with the final customer's product demand price elasticity decreases, the manufacturer's product quality level and wholesale price will increase, and the retailer's product retail price will also increase. In comparison with the direct channel scenario, the product quality level and the retail price will be much higher.

**Proof.** In this paper, we still use the backward induction method to solve the model. Thus, we use the first-order optimal condition with respect to $p_R$ and $λ_r$ in formula (16), which yields the following:

$$(α - β_R p_R / q_R)(1 - e^{-λ_r}) + (p_R - w)(-β_R / q_R)(1 - e^{-λ_r}) = 0 \quad (17)$$

$$(p_R - w)(α - β_R p_R / q_R)e^{-λ_r} - η_r = 0 \quad (18)$$

Therefore, we derive that

$$p_R = (αq_R + β_R w) / 2β_R, \quad λ_r = ln(p_R - w)(α - β_R p_R q_R^{-1}) / η_r \quad (19)$$

We substitute Eq (19) into formula (15) and use the first-order optimal condition with respect to $w$, and we obtain that

$$MaxΠ_M^R(q_R, λ_m, w) = (w - kq_R^2 / 2)(α / 2 - β_R w / 2q_R)(1 - e^{-λ_m}) - η_m λ_m = 0 \quad (20)$$

$$∂Π_M^R(q_R, λ_m, w) / ∂w = 0, \quad w^R = αq_R / 2β_R + kq_R^2 / 4 \quad (21)$$

We substitute Eq (21) into formula (20) and use the first-order optimal condition with respect to $λ_m$, which yields the following:

$$MaxΠ_M^R(q_R, λ_m) = (αq_R / 2β_R - kq_R^2 / 4)(α / 4 - kβ_R q_R / 8)(1 - e^{-λ_m}) - η_m λ_m = 0 \quad (22)$$

$$∂Π_M^R(q_R, λ_m) / ∂λ_m = 0, \quad λ_m^R = \ln(αq_R / 2β_R - kq_R^2 / 4)(α / 4 - kβ_R q_R / 8) / η_m \quad (23)$$

Thereafter, we use first-order and second-order optimal conditions with respect to $q_R$ in formula (22), which yields the following:

$$∂Π_M^R(q_R) / ∂q_R = (α / 2β_R - kq_R / 2)(α / 4 - kβ_R q_R / 8)(1 - e^{-λ_m})$$

$$-(αq_R / 2β_R - kq_R^2 / 4)(-kβ_R / 8)(1 - e^{-λ_m}) = 0 \quad (24)$$

$$∴q_R = 2α / 3kβ_R \text{ or } q_R = 2α / kβ_R$$

$$∵∂^2Π_M^R(q_R) / ∂q_R^2 < 0 \text{ (concave function)}$$

$$∴q_R < 4α / 3kβ_R$$

Therefore, we obtain that

$$q_R^* = 2\alpha \,/\, 3k\beta_R \tag{25}$$

$$\partial q_R^* \,/\, \partial\beta_R = -2\alpha \,/\, 3k\beta_R^2 < 0 \;(\text{decreasing function})$$

We substitute Eq (25) into formula (21) and (23) and rearrange as follows:

$$w^{R*} = 4\alpha^2 \,/\, 9k\beta_R^2, \;\; \lambda_m^{R*} = \ln\alpha^3 \,/\, 27k\eta_m\beta_R^2 \tag{26}$$

Therefore, we substitute Eqs (25) and (26) into formula (19) and rearrange as follows:

$$p_R^* = 5\alpha^2 \,/\, 9k\beta_R^2, \;\; \lambda_r^{R*} = \ln\alpha^3 \,/\, 54k\eta_r\beta_R^2 \tag{27}$$

$$\partial w^{R*} \,/\, \partial\beta_R = -8\alpha^2 \,/\, 9k\beta_R^3 < 0, \; \partial p_R^* \,/\, \partial\beta_R = -10\alpha^2 \,/\, 9k\beta_R^3 < 0$$

Based on the assumption condition and the Y.C Liu (2011) and Salma Karray (2015) research results, we assume $\beta_D = \varepsilon\beta_R$, where $\varepsilon$ is the demand elasticity ratio in a different distribution channel condition and $\varepsilon > 1$. The demand price elasticity for the direct channel will be greater than for the retail channel; i.e., the final customers are more sensitive to price in the direct channel. Samar K.M (2008) earlier had proved that $\eta_m = \eta_r$.

We compare Eqs (25), (26) and (27) with Eqs (11) and (12), respectively, as follows:

$$\because q_R^* - q_D^* > 0, \;\; \therefore q_R^* > q_D^*$$

$$\because p_R^* - w^{R*} > 0, \;\; w^{R*} - p_D^* > 0 \text{ and } p_R^* - p_D^* > 0, \;\; \therefore p_R^* > w^{R*} > p_D^*$$

**QED**

Based on proposition 2, we conclude that the manufacturer's product quality level in the retail channel will be much higher than that in the direct channel scenario; the retailer's retail price will be greater than the wholesale price, which will be also much higher than the manufacturer's sales price in the direct channel scenario.

**Corollary 2.1** $\lambda_m^{R*} > \lambda_r^{R*} > \lambda_m^{D*} \; (\varepsilon > 2)$.

**Proof.** We compare the manufacturer's quality prevention effort level and the retailer's quality inspection effort level in the retail channel with that in the direct channel; then, we determine that

$$\because \lambda_m^{R*} - \lambda_m^{D*} = \ln\alpha^3 \,/\, 27k\eta_m\beta_R^2 - \ln 2\alpha^3 \,/\, 27k\eta_m\beta_D^2 = \ln\beta_D^2 \,/\, 2\beta_R^2 = \ln\varepsilon^2 \,/\, 2 > 0(\varepsilon > \sqrt{2})$$

$$\lambda_m^{R*} - \lambda_m^{R*} = \ln\alpha^3 \,/\, 27k\eta_m\beta_R^2 - \ln\alpha^3 \,/\, 54k\eta_r\beta_R^2 = \ln 2\eta_r \,/\, \eta_m > 0$$

$$\lambda_r^{R*} - \lambda_m^{D*} = \ln\alpha^3 \,/\, 54k\eta_r\beta_R^2 - \ln 2\alpha^3 \,/\, 27k\eta_m\beta_D^2$$

$$= \ln\beta_D^2\eta_m \,/\, 4\beta_R^2\eta_r = \ln\varepsilon^2\eta_m \,/\, 4\eta_r > 0(\varepsilon > 2)$$

We conclude that $\lambda_m^{R*} > \lambda_r^{R*} > \lambda_m^{D*}(\varepsilon > 2)$

**QED**

Based on corollary 2.1, we can infer that the manufacturer's quality prevention effort level in the retail channel will be greater than the retailer's quality inspection effort level, which will also be greater than the manufacturer's quality prevention effort level in the direct channel.

**Corollary 2.2** $\prod_M^{R*}(q_R^*) > \prod_M^{D*}(q_D^*)$.

**Proof.** We substitute formula (25), (26) and (27) into Eqs (15) and (16); then, we find that

$$\therefore \prod_M^{R*} = 2\alpha^3 / 27k\beta_R^2 - \eta_m(1 + \ln \alpha^3 / 27k\eta_m\beta_R^2) \tag{28}$$

$$\prod_R^{R*} = \alpha^3 / 54k\beta_R^2 - \eta_r(1 + \ln \alpha^3 / 54k\eta_r\beta_R^2) \tag{29}$$

Therefore, we compare formula (28) with (29) to obtain that

$$\therefore \prod_M^{R*} - \prod_M^{D*} = 2\alpha^3(\beta_D^2 - \beta_R^2) / 27k\beta_D^2\beta_R^2 + \eta_m\ln 2\beta_R^2 / \beta_D^2 > 0$$

$$\therefore \text{we obtain that } \prod_M^{R*} > \prod_M^{D*}$$

**QED**

Based on corollary 2.2, we conclude that the manufacturer's expected profits in the retail channel will be greater than that in the direct channel scenario.

**Corollary 2.3.** $CS^{R*}(q^*_R, p^*_R) > CS^{D*}(q^*_D, p^*_D)$.

**Proof.** The final customer's consumer surplus in the retail channel will be described as follows

$$CS^{R*} = \int_a^b (\theta q_R^* - p_R^*)f(\theta)d\theta = (a+b)\alpha / 3k\beta_R - 5\alpha^2 / 9k\beta_R^2 \tag{30}$$

Therefore, we compare formula (30) with (14) to obtain

$$\because CS^{R*} - CS^{D*} = (a+b)\alpha(\beta_D - \beta_R) / 3k\beta_D\beta_R - \alpha^2(5\beta_D^2 - 4\beta_R^2) / 9k\beta_D^2\beta_R^2$$

We obtain $CS^{R*} > CS^{D*}$

**QED**

Based on corollary 2.3, we conclude that the final customer's consumer surplus in the retail channel will be greater than that in the direct channel.

## 6 Product quality strategy in mixed channel

In the mixed channel, the manufacturer may sell a product to the final customer directly through an online ordering system or sell wholesale to the retailer who will continue to sell the product to the final customer. Thereafter, the manufacturer will determine a transfer payment to the retailer to eliminate the channel conflict. Therefore, the manufacturer determines the product quality level, the quality prevention effort level, the wholesale price and the direct sale price, and the retailer determines the quality inspection level and the retail price. The

manufacturer and the retailer's stackelberg "leader-follower" control model can be described as follows:

$$Max\Pi_M^{MC}(q_{MC}, \lambda_m, w, P_D) = (p_D - kq_{MC}^2 / 2)(\alpha - \beta_D p_D / q_{MC})(1 - e^{-\lambda_m})$$

$$+(w - kq_{MC}^2 / 2)(\alpha - \beta_R p_R / q_{MC})(1 - e^{-\lambda_m}) - \eta_m \lambda_m - T \tag{31}$$

$$s.t.\{\lambda_r, p_R\} = arg\ Max\Pi_R^{MC}(\lambda_r, p_R)$$

$$Max\Pi_R^{MC}(\lambda_r, p_R) = (p_R - w)(\alpha - \beta_R p_R / q_{MC})(1 - e^{-\lambda_r}) - \eta_r \lambda_r + T \tag{32}$$

Therefore, formula (31) is the manufacturer's expected profits function, formula (32) is the retailer's expected profits function, and $T$ is the transfer payment.

**Proposition 3** In the mixed channel, in comparison with a direct channel and a retail channel scenario, the manufacturer's product quality level will be greater than which in the direct channel and less than which in the retail channel, i.e. $q_D^* < q_{MC}^* \leq q_R^*$, the wholesale price will decrease, i.e. $W^{MC*} \leq w^{R*}$, and the manufacturer's direct sale price will increase, i.e. $p_D^{MC*} > p_D^*$. In addition, the retailer's retail price will decrease, i.e. $p_R^{MC*} \leq p_R^*$.

**Proof.** We still use the backward induction method to solve the model. Thus, we use the first-order optimal condition with respect to $p_R$ and $\lambda_r$ in formula (32), which yields the following:

$$(\alpha - \beta_R p_R / q_{MC})(1 - e^{-\lambda_r}) + (p_R - w)(-\beta_R / q_{MC})(1 - e^{-\lambda_r}) = 0 \tag{33}$$

$$(p_R - w)(\alpha - \beta_R p_R / q_{MC})e^{-\lambda_r} - \eta_r = 0 \tag{34}$$

$$\because \partial^2 \Pi_R^{MC} / \partial p_R^2 = -2\beta_R(1 - e^{-\lambda_m}) / q_{MC} < 0 \ (\text{concave function}) \tag{35}$$

$$\therefore p_R{}^{MC} = (\alpha q_{MC} + \beta_R w) / 2\beta_R; \ \lambda_r{}^{MC} = \ln(p_R - w)(\alpha - \beta_R p_R q_{MC}^{-1}) / \eta_r \tag{36}$$

We substitute formula (36) into formula (31), which yields the following

$$Max\Pi_M^{MC}(q_{MC}, \lambda_m, w, P_D) = (p_D - 2^{-1}kq_{MC}^2)(\alpha - \beta_D p_D / q_{MC})(1 - e^{-\lambda_m})$$

$$+(w - 2^{-1}kq_{MC}^2)(\alpha / 2 - \beta_R w / 2q_{MC})(1 - e^{-\lambda_m}) - \eta_m \lambda_m - T \tag{37}$$

Therefore, we use the first-best condition $w$ and $p_D$ in formula (37) and obtain that

$$\therefore w^{MC} = \alpha q_{MC} / 2\beta_R + kq_{MC}^2 / 4, \ p_D^{MC} = \alpha q_{MC} / 2\beta_D + kq_{MC}^2 / 4 \tag{38}$$

Thereafter, we substitute formula (38) into formula (37), which yields the following

$$Max\Pi_M^{MC}(q_{MC}, \lambda_m) = (\alpha q_{MC} / 2\beta_D - 4^{-1}kq_{MC}^2)(\alpha / 2 - \beta_D kq_{MC} / 4)(1 - e^{-\lambda_m})$$

$$+(\alpha q_{MC} / 2\beta_R - 4^{-1}kq_{MC}^2)(\alpha / 4 - \beta_R kq_{MC} / 8)(1 - e^{-\lambda_m}) - \eta_m \lambda_m - T \tag{39}$$

We use the first-best condition $\lambda_m$ and first-best and second-best condition $q_{MC}$ and obtain

that

$$\lambda_m^{MC} = \ln \eta_m^{-1}[(\frac{\alpha q_{MC}}{2\beta_D} - \frac{kq_{MC}^2}{4})(\frac{\alpha}{2} - \frac{\beta_D kq_{MC}}{4}) + (\frac{\alpha q_{MC}}{2\beta_R} - \frac{kq_{MC}^2}{4})(\frac{\alpha}{4} - \frac{\beta_R kq_{MC}}{8})] \tag{40}$$

$$\because \partial \Pi_M^{MC}(q_{MC}) / \partial q_{MC} = 0, \quad \partial^2 \Pi_M^{MC}(q_{MC}) / \partial q_{MC}^2 < 0 \text{ (concave function)} \tag{41}$$

$$\therefore q_{MC}^* = 2\alpha(4 - m) / 3k(\beta_D + \beta_R), \quad m = (10 - 3(\varepsilon + \varepsilon^{-1}))^{1/2} \quad (2 < \varepsilon \le 3) \tag{42}$$

Therefore, we substitute formula (42) into formula (36), (38) and (40) and obtain that

$$\therefore w^{MC*} = \alpha^2(4 - m) / 3k\beta_R(\beta_D + \beta_R) + \alpha^2(4 - m)^2 / 9k(\beta_D + \beta_R)^2$$

$$\lambda_m^{MC*} = \ln \eta_m^{-1}[\alpha^3(4 - m) / 6k\beta_D\beta_R - \alpha^3(4 - m)^2(8 + m) / 54k(\beta_D + \beta_R)^2] \tag{43}$$

$$p_D^{MC*} = \alpha^2(4 - m) / 3k\beta_D(\beta_D + \beta_R) + \alpha^2(4 - m)^2 / 9k(\beta_D + \beta_R)^2$$

$$p_R^{MC*} = \alpha^2(4 - m) / 2k\beta_R(\beta_D + \beta_R) + \alpha^2(4 - m)^2 / 18k(\beta_D + \beta_R)^2$$

$$\lambda_r^{MC*} = \ln \eta_r^{-1}[\frac{\alpha^3(4 - m)}{24k\beta_R(\beta_D + \beta_R)} + \frac{\alpha^3(4 - m)^3\beta_R}{216k(\beta_D + \beta_R)^3} - \frac{\alpha^3(4 - m)^2}{36k(\beta_D + \beta_R)^2}] \tag{44}$$

We compare the product quality level, the wholesale price, the sales price and the retail price in the mixed channel, with which in the direct channel and the retail channel scenario yields the following:

$$\because q_{MC}^* / q_D^* = \beta_D(4 - m) / (\beta_D + \beta_R) > 1(2 < \varepsilon \le 3)$$

$$q_{MC}^* / q_R^* = \beta_R(4 - m) / (\beta_D + \beta_R) = \begin{cases} < 1, & 2 < \varepsilon < 3 \\ = 1, & \varepsilon = 3 \end{cases}$$

$$\therefore q_D^* < q_{MC}^* \le q_R^* \tag{45}$$

$$\therefore w^{MC*} - w^{R*} = \frac{\alpha^2(4 - m)}{3k(\beta_D + \beta_R)}(\frac{1}{\beta_R} + \frac{4 - m}{3(\beta_D + \beta_R)}) - \frac{4\alpha^2}{9k\beta_R^2} = \begin{cases} < 0, & 2 < \varepsilon < 3 \\ = 0, & \varepsilon = 3 \end{cases}$$

$$p_D^* - p_D^{MC*} = \frac{4\alpha^2}{9k\beta_D^2} - \frac{\alpha^2(4 - m)^2}{9k(\beta_D + \beta_R)^2} - \frac{\alpha^2(4 - m)}{3k\beta_D(\beta_D + \beta_R)} < 0$$

$$p_R^* - p_R^{MC*} = \frac{5\alpha^2}{9k\beta_R^2} - \frac{\alpha^2(4 - m)^2}{18k(\beta_D + \beta_R)^2} - \frac{\alpha^2(4 - m)}{2k\beta_R(\beta_D + \beta_R)} = \begin{cases} > 0, & 2 < \varepsilon < 3 \\ = 0, & \varepsilon = 3 \end{cases} \tag{46}$$

Then we can infer that $w^{MC*} \le w^{R*}$, $P_D^* < p_D^{MC*}$ and $p_R^{MC*} \le p_R^*$.

## QED

Based on proposition 3, we conclude that, in the mixed channel, the manufacturer's product quality level will be greater than which in the direct channel and less than which in the retail

channel. In addition, the wholesale price will decrease, the manufacturer's direct sale price will increase and the retailer's retail price will decrease.

**Corollary 3.1** $\lambda_m^{MC*} \geq \lambda_m^{R*} > \lambda_m^{D*}, \lambda_r^{MC*} \geq \lambda_r^{R*}$.

**Proof.** We compare the manufacturer's quality prevention effort level and the retailer's quality inspection effort level in the mixed channel with which in the direct channel and the retail channel scenario and obtain the following:

$$\lambda_m^{MC*} - \lambda_m^{R*} = \ln\left[\frac{9(4-m)\beta_R}{4\beta_D} - \frac{(4-m)^2(8+m)\beta_R^2}{4(\beta_D+\beta_R)^2}\right] = \begin{cases} > 0, & 2 < \varepsilon < 3 \\ = 0, & \varepsilon = 3 \end{cases} \quad (47)$$

$$\therefore \lambda_m^{MC*} \geq \lambda_m^{R*}$$

By Corollary 2.1, we obtain that $\lambda_m^{R*} > \lambda_m^{D*}$; then, we can infer that $\lambda_m^{MC*} \geq \lambda_m^{R*} > \lambda_m^{D*}$.

$$\because \lambda_r^{MC*} - \lambda_r^{R*} = \ln\left[\frac{27(4-m)\beta_R}{12(\beta_D-\beta_R)} + \frac{(4-m)^3\beta_R^3}{4(\beta_D+\beta_R)^3} - \frac{3(4-m)^2\beta_R^2}{2(\beta_D+\beta_R)^2}\right] = \begin{cases} > 0, & 2 < \varepsilon < 3 \\ = 0, & \varepsilon = 3 \end{cases} \quad (48)$$

$$\therefore \lambda_r^{MC*} \geq \lambda_r^{R*}$$

## QED

Based on corollary 3.1, we conclude that, in the mixed channel, the manufacturer's quality prevention effort level will be greater than that in the retail channel and the direct channel, and the retailer's quality inspection effort level will be greater than that in the retail channel, which will effectively eliminate the "free-riding behavior.

**Corollary 3.2** $\prod_M^{R*}(q_R^*) > \prod_M^{MC*}(q_{MC}^*) > \prod_M^{D*}(q_D^*), \prod_R^{MC*}(q_{MC}^*) > \prod_R^{R*}(q_R^*)$.

**Proof.** We substitute formula (42), (43) and (44) into Eqs (31) and (32), which yields the following:

$$\prod_M^{MC*} = \left(\frac{\alpha^3(4-m)}{6k\beta_D\beta_R} - \frac{\alpha^3(4-m)^2(8+m)}{54k(\beta_D+\beta_R)^2}\right) - \eta_m\ln\eta_m^{-1}\left[\frac{\alpha^3(4-m)}{6k\beta_D\beta_R} - \frac{\alpha^3(4-m)^2(8+m)}{54k(\beta_D+\beta_R)^2}\right]$$
$$- \eta_m - T \quad (49)$$

$$\prod_R^{MC*} = \left(\frac{\alpha^3(4-m)}{24k\beta_R(\beta_D+\beta_R)} + \frac{\alpha^3(4-m)^3\beta_R}{216k(\beta_D+\beta_R)^3} - \frac{\alpha^3(4-m)^2}{36k(\beta_D+\beta_R)^2}\right)$$

$$-\eta_r\ln\eta_r^{-1}\left[\frac{\alpha^3(4-m)}{24k\beta_R(\beta_D+\beta_R)} + \frac{\alpha^3(4-m)^3\beta_R}{216k(\beta_D+\beta_R)^3} - \frac{\alpha^3(4-m)^2}{36k(\beta_D+\beta_R)^2}\right] - \eta_r + T \quad (50)$$

Therefore, we compare formula (49) and (50) with formula (13), (28) and (29) and obtain that

$$\therefore \prod_M^{MC*}(q_{MC}^*) - \prod_M^{D*}(q_D^*) > 0, \ \prod_M^{MC*}(q_{MC}^*) - \prod_M^{R*}(q_R^*) < 0, \ \prod_R^{MC*}(q_{MC}^*) - \prod_R^{R*}(q_R^*) > 0$$

$$\therefore \prod_M^{R*}(q_R^*) > \prod_M^{MC*}(q_{MC}^*) > \prod_M^{D*}(q_D^*), \ \prod_R^{MC*}(q_{MC}^*) > \prod_R^{R*}(q_R^*) \quad (51)$$

**QED**

Based on corollary 3.2, we infer that, in the mixed channel, the manufacturer's expected profits will be less than that in the retail channel but will be greater than the profits in the direct channel; additionally, the retailer's expected profits will be greater than that in the retail channel.

**Corollary 3.3** $CS^{MC*}(q_{MC}^*) > CS^{R*}(q_R^*) > CS^{D*}(q_D^*)$.

**Proof.** The final customer's consumer surplus in a mixed channel will be described as follows

$$CS^{MC*} = \int_a^{(a+b)/2} (\theta q_{MC}^* - p_D^{MC*})f(\theta)\,d\theta + \int_{(a+b)/2}^b (\theta q_{MC}^* - p_R^{MC*})f(\theta)\,d\theta$$

$$= \frac{(a-b)(4-m)\alpha}{3k(\beta_D + \beta_R)} - \frac{1}{2}\Big[\frac{\alpha^2(4-m)^2}{6k(\beta_D + \beta R)^2} + \frac{\alpha^2(4-m)(3\beta_D + 2\beta_R)}{6k\beta_D\beta_R(\beta_D + \beta_R)}\Big] \tag{52}$$

Thereafter, we compare formula (52) with formula (14) and (30) and obtain that

$$\therefore CS^{MC*} - CS^{D*} > 0, \quad CS^{MC*} - CS^{R*} > 0, \quad CS^{R*} > CS^{D*} \text{ (corollary 2.3 had proved)}$$

We find that $CS^{MC*} > CS^{R*} > CS^{D*}$.

**QED**

Based on corollary 3.3, we can infer that, in the mixed channel, the final customer's consumer surplus will be greater than that in the retail channel and the direct channel.

## 7 Numerical example

In this paper, we assume a manufacturer that sells computer components through a retailer (retail channel) or internet online system (direct channel) or through a mixed channel to the final customer. The parameters are described as follows: $k = 2$, $\eta_m = \eta_r = 1$, $\alpha = 60$, $\theta \sim U(0,60)$, $T = 120$, $\varepsilon = \{2.5, 3.0\}$, $\beta_R \sim [2.5, 3.5]$. We use numerical computing by Matlab 7.0 and obtain the results, as shown in Tables 1–4.

Based on Table 1, we conclude that, in the direct channel, the manufacturer's product quality level, the quality prevention effort level and the direct sale price will increase, and the

**Table 1. In the direct channel scenario.**

| | | | $\varepsilon = 3.0$ | | | | | | $\varepsilon = 2.5$ | | |
| --- | --- | --- | --- | --- | --- | --- | --- | --- | --- | --- | --- |
| $\beta_D$ | $\lambda_m^{D*}$ | $q_D^*$ | $p_D^*$ | $\prod_M^{D*}$ | $CS^{D*}$ | $\beta_D$ | $\lambda_m^{D*}$ | $q_D^*$ | $p_D^*$ | $\prod_M^{D*}$ | $CS^{D*}$ |
| 7.50 | 4.9574 | 2.6680 | 14.2224 | 136.2650 | 65.7776 | 6.25 | 5.3220 | 3.2000 | 20.4800 | 198.4780 | 75.5200 |
| 7.80 | 4.8790 | 2.5654 | 13.1494 | 125.6137 | 63.7737 | 6.50 | 5.2436 | 3.0769 | 18.9349 | 183.1055 | 73.3728 |
| 8.10 | 4.8035 | 2.4704 | 12.1934 | 116.1293 | 61.8807 | 6.75 | 5.1681 | 2.9630 | 17.5583 | 169.4149 | 71.3306 |
| 8.40 | 4.7307 | 2.3821 | 11.3380 | 107.6481 | 60.0906 | 7.00 | 5.0954 | 2.8571 | 16.3265 | 157.1699 | 69.3878 |
| 8.70 | 4.6606 | 2.3000 | 10.5696 | 100.0339 | 58.3960 | 7.25 | 5.0252 | 2.7586 | 15.2200 | 146.1746 | 67.5386 |
| 9.00 | 4.5927 | 2.2233 | 9.8767 | 93.1728 | 56.7900 | 7.50 | 4.9574 | 2.6667 | 14.2222 | 136.2648 | 65.7778 |
| 9.30 | 4.5272 | 2.1516 | 9.2497 | 86.9692 | 55.2664 | 7.75 | 4.8918 | 2.5806 | 13.3195 | 127.3028 | 64.0999 |
| 9.60 | 4.4637 | 2.0844 | 8.6807 | 81.3420 | 53.8193 | 8.00 | 4.8283 | 2.5000 | 12.5000 | 119.1717 | 62.5000 |
| 9.90 | 4.4021 | 2.0212 | 8.1625 | 76.2223 | 52.4435 | 8.25 | 4.7668 | 2.4242 | 11.7539 | 111.7723 | 60.9734 |
| 10.2000 | 4.3424 | 1.9618 | 7.6894 | 71.5512 | 51.1341 | 8.50 | 4.7071 | 2.3529 | 11.0727 | 105.0196 | 59.5156 |
| 10.5000 | 4.2844 | 1.9057 | 7.2563 | 67.2780 | 49.8865 | 8.75 | 4.6491 | 2.2857 | 10.4490 | 98.8407 | 58.1224 |

**Table 2. In the retail channel scenario.**

| $B_R$ | $\lambda_m^{R*}$ | $\lambda_r^{R*}$ | $q_R^*$ | $w_R^*$ | $p_R^*$ | $\prod_M^{R*}$ | $\prod_R^{R*}$ | $CS^{R*}$ |
|---|---|---|---|---|---|---|---|---|
| 2.50 | 6.4615 | 5.7683 | 8.0000 | 128.0000 | 160.0000 | 1272.5385 | 313.2317 | 80.0000 |
| 2.60 | 6.3830 | 5.6899 | 7.6923 | 118.3432 | 147.9290 | 1176.0489 | 289.1681 | 82.8402 |
| 2.70 | 6.3075 | 5.6144 | 7.4074 | 109.7394 | 137.1742 | 1090.0861 | 267.7340 | 85.0480 |
| 2.80 | 6.2348 | 5.5417 | 7.1429 | 102.0408 | 127.5510 | 1013.1734 | 248.5604 | 86.7347 |
| 2.90 | 6.1646 | 5.4715 | 6.8966 | 95.1249 | 118.9061 | 944.0839 | 231.3406 | 87.9905 |
| 3.00 | 6.0968 | 5.4037 | 6.6667 | 88.8889 | 111.1111 | 881.7921 | 215.8185 | 88.8889 |
| 3.10 | 6.0312 | 5.3381 | 6.4516 | 83.2466 | 104.0583 | 825.4349 | 201.7784 | 89.4901 |
| 3.20 | 5.9677 | 5.2746 | 6.2500 | 78.1250 | 97.6563 | 774.2823 | 189.0379 | 89.8438 |
| 3.30 | 5.9062 | 5.2131 | 6.0606 | 73.4619 | 91.8274 | 727.7127 | 177.4417 | 89.9908 |
| 3.40 | 5.8465 | 5.1534 | 5.8824 | 69.2042 | 86.5052 | 685.1950 | 166.8570 | 89.9654 |
| 3.50 | 5.7885 | 5.0954 | 5.7143 | 65.3061 | 81.6327 | 646.2727 | 157.1699 | 89.7959 |

customer's consumer surplus will also increase with the decreasing in the product demand price elasticity, which will benefit the manufacturer and the final customer when the distribution channel demand elasticity ratio decreases.

Based on Table 2, we can infer that, in the retail channel, the final customer's consumer surplus will increase with the increasing in product demand price elasticity. Compared with the direct channel, the manufacturer's product quality level, the quality prevention effort level and the expected profits will increase, and the customer's consumer surplus will also increase.

Based on Tables 3 and 4 and Figs 2 and 3, we conclude that in the mixed channel compared with that in the direct channel and the retail channel, the manufacturer will determine a transfer payment to eliminate channel conflict, the manufacturer's quality prevention effort level and retailer's quality inspection effort level will increase, which will effectively eliminate the "free-riding behavior", the manufacturer's expected profits will be higher than which in the direct channel and less than which in the retail channel; in addition, the retailer's expected profits and the final consumer surplus will increase.

## 8 Conclusions and future research

In this paper, we construct a product quality control model of three types of distribution channel (direct channel, retail channel and mixed channel) in a three-echelon supply chain, which

**Table 3. In the mixed channel scenario ($\varepsilon = 3.0$).**

| $\beta_R$ | $\lambda_m^{MC*}$ | $\lambda_r^{MC*}$ | $q_{MC}^*$ | $w_{MC}^*$ | $p_D^{MC*}$ | $p_R^{MC*}$ | $\prod_M^{MC*}$ | $\prod_R^{MC*}$ | $CS^{MC*}$ |
|---|---|---|---|---|---|---|---|---|---|
| 2.50 | 7.1546 | 5.7683 | 8.0000 | 128.0000 | 64.0000 | 160.0000 | 1151.8454 | 433.2317 | 128.0000 |
| 2.60 | 7.0762 | 5.6899 | 7.6923 | 118.3432 | 59.1716 | 147.9290 | 1055.3558 | 409.1681 | 127.2189 |
| 2.70 | 7.0007 | 5.6144 | 7.4074 | 109.7394 | 54.8697 | 137.1742 | 969.3930 | 387.7340 | 126.2003 |
| 2.80 | 6.9280 | 5.5417 | 7.1429 | 102.0408 | 51.0204 | 127.5510 | 892.4802 | 368.5604 | 125.0000 |
| 2.90 | 6.8578 | 5.4715 | 6.8966 | 95.1249 | 47.5624 | 118.9061 | 823.3907 | 351.3406 | 123.6623 |
| 3.00 | 6.7900 | 5.4037 | 6.6667 | 88.8889 | 44.4444 | 111.1111 | 761.0989 | 335.8185 | 122.2222 |
| 3.10 | 6.7244 | 5.3381 | 6.4516 | 83.2466 | 41.6233 | 104.0583 | 704.7418 | 321.7784 | 120.7076 |
| 3.20 | 6.6609 | 5.2746 | 6.2500 | 78.1250 | 39.0625 | 97.6563 | 653.5891 | 309.0379 | 119.1406 |
| 3.30 | 6.5994 | 5.2131 | 6.0606 | 73.4619 | 36.7309 | 91.8274 | 607.0196 | 297.4417 | 117.5390 |
| 3.40 | 6.5396 | 5.1534 | 5.8824 | 69.2042 | 34.6021 | 86.5052 | 564.5019 | 286.8570 | 115.9170 |
| 3.50 | 6.4817 | 5.0954 | 5.7143 | 65.3061 | 32.6531 | 81.6327 | 525.5796 | 277.1699 | 114.2857 |

**Table 4. In the mixed channel scenario ($\varepsilon = 2.5$).**

| $\beta_R$ | $\lambda_m^{MC*}$ | $\lambda_r^{MC*}$ | $q_{MC}^*$ | $w_{MC}^*$ | $p_D^{MC*}$ | $p_R^{MC*}$ | $\prod_M^{MC*}$ | $\prod_R^{MC*}$ | $CS^{MC*}$ |
|---|---|---|---|---|---|---|---|---|---|
| 2.50 | 7.2017 | 5.7914 | 6.5360 | 99.8128 | 52.7453 | 128.3522 | 1213.5631 | 440.6938 | 105.5656 |
| 2.60 | 7.1233 | 5.7130 | 6.2846 | 92.2825 | 48.7660 | 118.6688 | 1112.4137 | 416.0655 | 104.8541 |
| 2.70 | 7.0478 | 5.6375 | 6.0519 | 85.5734 | 45.2206 | 110.0413 | 1022.2992 | 394.1283 | 103.9564 |
| 2.80 | 6.9751 | 5.5648 | 5.8357 | 79.5702 | 42.0482 | 102.3216 | 941.6716 | 374.5045 | 102.9172 |
| 2.90 | 6.9049 | 5.4946 | 5.6345 | 74.1772 | 39.1983 | 95.3866 | 869.2449 | 356.8803 | 101.7716 |
| 3.00 | 6.8371 | 5.4268 | 5.4467 | 69.3144 | 36.6287 | 89.1334 | 803.9440 | 340.9935 | 100.5476 |
| 3.10 | 6.7715 | 5.3612 | 5.2710 | 64.9147 | 34.3036 | 83.4757 | 744.8643 | 326.6235 | 99.2671 |
| 3.20 | 6.7080 | 5.2977 | 5.1063 | 60.9209 | 32.1932 | 78.3399 | 691.2402 | 313.5834 | 97.9478 |
| 3.30 | 6.6465 | 5.2362 | 4.9515 | 57.2847 | 30.2716 | 73.6640 | 642.4206 | 301.7145 | 96.6037 |
| 3.40 | 6.5868 | 5.1765 | 4.8059 | 53.9645 | 28.5171 | 69.3946 | 597.8483 | 290.8809 | 95.2459 |
| 3.50 | 6.5288 | 5.1185 | 4.6686 | 50.9249 | 26.9109 | 65.4858 | 557.0451 | 280.9658 | 93.8833 |

is comprised by one manufacturer, one retailer and the final customer, and then we discuss how to design a distribution channel strategy and craft a quality control strategy. Furthermore, our paper analyzes three types of distribution channel strategies regarding how to influence the manufacturer's product quality decision and quality prevention strategy, the retailer's product pricing decision and quality inspection strategy, and the final customer's product demand decision. We compare the product quality level in three types of distribution channels and solve the manufacturer's and retailer's expected profits functions and the final customer's consumer surplus. In addition, we introduce the distribution channel demand elasticity ratio to analyze the influence of determining the product quality control strategy.

Our paper demonstrates that, in the direct channel, the manufacturer's product quality level, the quality prevention effort level and the direct sale price will increase, and the

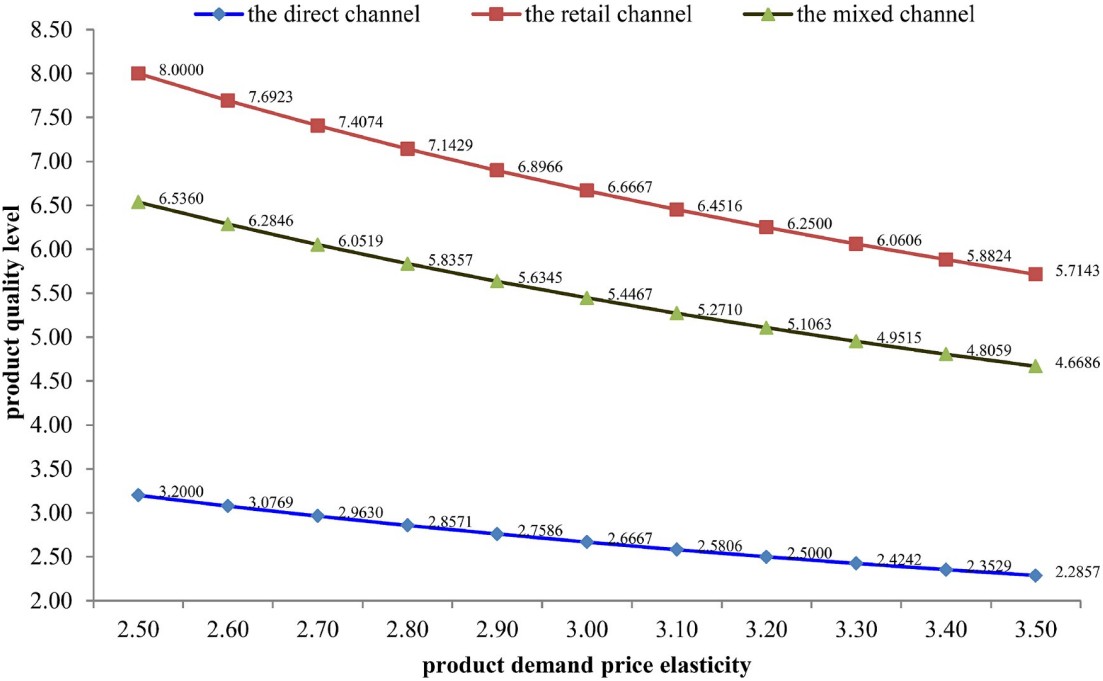

**Fig 2. The product quality level in three types of distribution channels.**

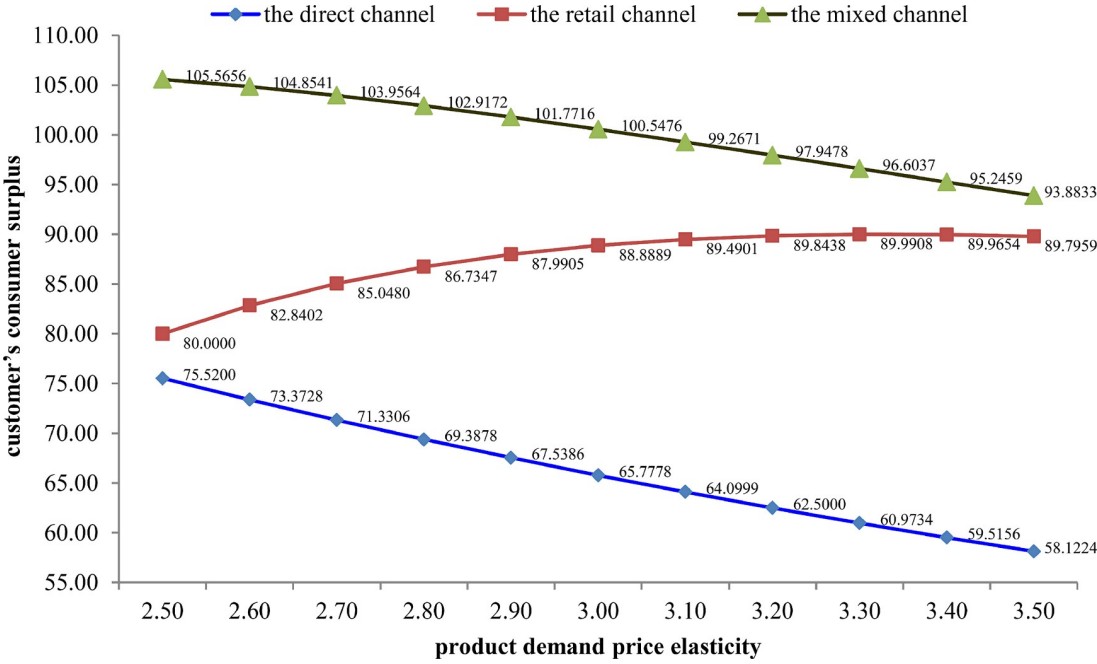

**Fig 3. The customer's consumer surplus in three types of distribution channels.**

customer's consumer surplus will also increase with the decreasing in the products demand price elasticity. In addition, in the retail channel which is compared with the direct channel scenario, the manufacturer's product quality level, the wholesale price, the quality prevention effort level and expected profits will increase, and the retailer's retail price, the quality inspection effort level and the customer's consumer surplus will be much higher. In the mixed channel, the manufacturer will determine the transfer payment to eliminate channel conflict, the manufacturer's quality prevention effort level and the retailer's quality inspection effort level will increase, which will effectively eliminate the "free-riding behavior". In addition, the manufacturer's expected profits will be higher than that in the direct channel and less than that in the retail channel. The retailer's expected profits and the final consumer surplus will also increase, and our conclusions will be a strong complement to the research field. Most importantly, we conduct a numerical sample analysis that demonstrates the model's effectiveness and the conclusions' correctness and will also indicate a specific application in practice.

In further research, we will assume that the manufacturer's quality prevention effort level and the retailer's quality inspection effort level have incomplete information regarding how to craft a product quality control strategy in three types of distribution channels; then, we will also attempt to construct a multi-stage, repeat and asymmetry information dynamic game model and analyze the distribution channel strategy regarding how to influence the manufacturer's and retailer's expected profits function, the final customer's consumer surplus and social welfare.

## Acknowledgments

The authors are grateful to the referees for their valuable comments and their helps on how to improve the quality of our paper.

## Author Contributions

**Writing – original draft:** Lilong Zhu.

**Writing – review & editing:** Lilong Zhu.

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
