## [Decision Letter · Decision Letter 0]

8 Jan 2020

PONE-D-19-20228

Supply Chain Product Quality Control Strategy in Three Types of Distribution Channels

PLOS ONE

Dear Authors,

Thank you for submitting your manuscript to PLOS ONE. After careful consideration, we feel that it has merit but does not fully meet PLOS ONE’s publication criteria as it currently stands. Therefore, we invite you to submit a revised version of the manuscript that addresses the points raised during the review process.

We would appreciate receiving your revised manuscript by 30.1.2020. To enhance the reproducibility of your results, we recommend that if applicable you deposit your laboratory protocols in protocols.io, where a protocol can be assigned its own identifier (DOI) such that it can be cited independently in the future. For instructions see: http://journals.plos.org/plosone/s/submission-guidelines#loc-laboratory-protocols

We look forward to receiving your revised manuscript.

Kind regards,

Dejan Dragan, PhD

Academic Editor

PLOS ONE

Additional Editor Comments (if provided):

Editor's initial comments to the paper:

Supply Chain Product Quality Control Strategy in Three Types of Distribution Channels

The paper deploys a three-stage Stackelberg dynamic game analysis and constructs a product quality control strategy model for three types of distribution channels (direct channel, retail channel, and mixed channel) in a three-echelon supply chain, which is composed of one manufacturer, one retailer and the final customer. Furthermore, the paper studies how to design a distribution channel strategy and provides a product quality control strategy. Here, three types of distribution channels strategy in the context of how they influence a manufacturer’s product quality decision and prevention strategy, a retailer’s product pricing decision and quality inspection strategy, and the final customer’s product demand decision, are analyzed. The subject of this research is up-to-date and fundamentally interesting for scholars from the field of SCM and OR.

The paper, in general, roughly satisfies all rigor requirements that are demanded from Plos One. The red clue remains more or less consistent all over the paper, while the derivations of equations seem to be adequate.

However, the editor has detected some major issues based on his own perception, the review of the additional reviewer, and the Plos One criteria for the papers, which should be corrected prior to the further publishing process.

Comments of the AE:

1. It is not clear enough emphasized what the main contribution of the paper is, i.e., what has been done new, if compared with the work of the other researchers. Here, the borderline should be clearly highlighted, while the main contributions (novelties, originality) should be more explicitly given in the introduction and other places, where necessary.

2. Figure 1 should be improved in the sense of informative content meaning that the reader immediately understands the main point without even looking at the corresponding text.

Comments regarding the Plos One criteria for the papers:

There exists a certain doubt whether the paper meets PLOS ONE criteria for papers that describe new methods or software for applications. Specifically, these reports must meet the criteria of utility, validation, and availability, which are described in detail at http://journals.plos.org/plosone/s/submission-guidelines#loc-methods-software-databases-and-tools. On this basis, the AE warns the authors to reconsider these facts and give the rationale and/or explanation fortified with the facts, let be in the paper itself or in the answers to reviewers or cover letter.

Comments of Reviewer #1

Summary (see reviewer 1 comments.pdf):

Additional comments are available in the file: reviewer 1 comments.pdf.

Journal Requirements:

2. Thank you for stating the following in the Acknowledgments Section of your manuscript: "This work was supported by the Humanities and Social Sciences Foundation of the Ministry of Education in China under grant No.17YJA630147, Nature Science Foundation of Shandong Province under grant No.ZR2019MG017 and National Social Science Foundation of China under grant No.13AGL012."

3. Please include your tables as part of your main manuscript and remove the individual files. Please note that supplementary tables should be uploaded as separate "supporting information" files.

Reviewers' comments:

Reviewer's Responses to Questions

**Comments to the Author**

1. Is the manuscript technically sound, and do the data support the conclusions?

Reviewer #1: Partly

2. Has the statistical analysis been performed appropriately and rigorously? 

Reviewer #1: Yes

3. Have the authors made all data underlying the findings in their manuscript fully available?

Reviewer #1: Yes

4. Is the manuscript presented in an intelligible fashion and written in standard English?

Reviewer #1: No

5. Review Comments to the Author

Reviewer #1: 1. The language of the article should be more concise and accurate.

2. It is better to complement the differences between this study and the existing research literature.

3. The conclusions of the study for the the objectives mentioned in the abstract as ‘This paper studies how to design a distribution channel strategy and provides a product quality control strategy’ are not very clear.

4. In this article, we can not see how consumer utility and consumer behavior affect product market demand.

5. In Mixed Channel, how does the product demand between the two channels affect each other?

6. PLOS authors have the option to publish the peer review history of their article (what does this mean?). If published, this will include your full peer review and any attached files.

Reviewer #1: No

---

## [Author Response · Author response to Decision Letter 0]

2 Feb 2020

Responses to Editors and Reviewers

Dear Professor Dejan Dragan and Reviewers,

Thank you very much for your suggestions and critical comments about our paper submitted to PLOS ONE (Manuscript ID: PONE-D-19-20228). The revised title is “Supply Chain Product Quality Control Strategy in Three Types of Distribution Channels”.

We are also thankful to the reviewers for their critical reading and valuable comments on the manuscript. Those comments were very helpful for providing direction for our further studies. We have tried our best to revise our manuscript according to the comments. Attached, please find the revised version, which we would like to resubmit for your kind consideration. The main revised parts are marked in blue in the paper. The following is a detailed explanation how we have complied with the editor’s and reviewers’ suggestions.

Responds to the editor’s comments:

Comment #1:

The paper deploys a three-stage Stackelberg dynamic game analysis and constructs a product quality control strategy model for three types of distribution channels (direct channel, retail channel, and mixed channel) in a three-echelon supply chain, which is composed of one manufacturer, one retailer and the final customer. Furthermore, the paper studies how to design a distribution channel strategy and provides a product quality control strategy. Here, three types of distribution channels strategy in the context of how they influence a manufacturer’s product quality decision and prevention strategy, a retailer’s product pricing decision and quality inspection strategy, and the final customer’s product demand decision, are analyzed. The subject of this research is up-to-date and fundamentally interesting for scholars from the field of SCM and OR.

The paper, in general, roughly satisfies all rigor requirements that are demanded from Plos One. The red clue remains more or less consistent all over the paper, while the derivations of equations seem to be adequate.

Response: Thank you very much for the highly praises and the valuable suggestions. Our paper constructs a product quality control strategy model for three types of distribution channels (direct channel, retail channel and mixed channel) in a three-echelon supply chain which use a three-stage stackelberg dynamic game analysis, and then, analyzes three types of distribution channels strategy in the context of how they influence a manufacturer’s product quality decision and quality prevention strategy, a retailer’s product pricing decision and quality inspection strategy, and the final customer’s product demand decision. Most importantly, we conduct a numerical sample analysis that will prove the model’s effectiveness and indicate a specific application in practice.

Comment #2: 

It is not clear enough emphasized what the main contribution of the paper is, i.e., what has been done new, if compared with the work of the other researchers. Here, the borderline should be clearly highlighted, while the main contributions (novelties, originality) should be more explicitly given in the introduction and other places, where necessary.

Response: Thank you very much for the valuable suggestions. We rewrite and emphasize the paper’s main contributions in every paragraph in “2 Related Literature”, just like: (1) Our paper differs from the existing literature by introducing the distribution channel demand elasticity ratio and investigating how to construct a product quality control strategy model and channel coordination in three types of distribution channels (direct channel, retail channel, and mixed channel) by providing a new perspective. (2) Our model contributes to the product quality control strategy research by constructing a distribution channel model in a three-echelon supply chain which is composed of one manufacturer, one retailer and the final customer based on a three-stage stackelberg dynamic game. Then, the model considers the manufacturer’s product quality decision and quality prevention strategy, the retailer’s product pricing decision and quality inspection strategy, and the final customer’s product demand decision. (3) we also establish a product quality control strategy model in three types of distribution channels to eliminate the influence of “channel conflict” and “free-riding behavior”, which will improve the manufacturer’s and retailer’s expected profits and the final customer’s consumer surplus.

Comment #3: 

Figure 1 should be improved in the sense of informative content meaning that the reader immediately understands the main point without even looking at the corresponding text.

Response: Thank you very much for the valuable suggestions. We redraw the Figure 1 Three types of distribution channels decision system which improve in the sense of informative content meaning that the reader immediately understands the main point of our paper.

Figure 1 Three types of distribution channels decision system

Responds to the reviewer’s comments:

Comment #1:

The language of the article should be more concise and accurate.

Response: Thank you very much for the valuable suggestions. We made necessary revisions and language editing in the manuscript according to your suggestions, and the English has also been edited by Wiley English Language Editing Services.

Comment #2:

It is better to complement the differences between this study and the existing research literature.

Response: Thank you very much for the valuable suggestions. We rewrite and emphasize the paper’s main contributions in every paragraph in “2 Related Literature”, just like: (1) Our paper differs from the existing literature by introducing the distribution channel demand elasticity ratio and investigating how to construct a product quality control strategy model and channel coordination in three types of distribution channels (direct channel, retail channel, and mixed channel) by providing a new perspective. (2) Our model contributes to the product quality control strategy research by constructing a distribution channel model in a three-echelon supply chain which is composed of one manufacturer, one retailer and the final customer based on a three-stage stackelberg dynamic game. Then, the model considers the manufacturer’s product quality decision and quality prevention strategy, the retailer’s product pricing decision and quality inspection strategy, and the final customer’s product demand decision. (3) we also establish a product quality control strategy model in three types of distribution channels to eliminate the influence of “channel conflict” and “free-riding behavior”, which will improve the manufacturer’s and retailer’s expected profits and the final customer’s consumer surplus.

Comment #3: 

The conclusions of the study for the objectives mentioned in the abstract as ‘This paper studies how to design a distribution channel strategy and provides a product quality control strategy’ are not very clear.

Response: Thank you very much for the valuable suggestions. With the changing in customer or consumer buying behavior, more and more companies are beginning to redesign or rebuild their distribution channel structure, Such as HP, Nike, Lenovo, in addition to focus on the traditional retail channel, have also opened up an internet channel(direct channel); Dell, MI has been focused on internet channel in the past, and now also began selling products in traditional retail channel; and Apple, Haier sell their products in the traditional retail channel and internet channel in the same time, which used a mixed channel structure. In our paper, we will construct a distribution channel strategy model in a three echelon supply chain that is composed of one manufacturer, one retailer and a final customer based on a three-stage stackelberg dynamic game. Furthermore, we will introduce the distribution channel demand elasticity ratio and investigate how to craft a product quality control strategy in three types of distribution channels (direct channel, retail channel, and mixed channel), Most important, we will determine the manufacturer’s product quality level, quality prevention effort level, wholesale price, direct sale price, and the retailer’s product quality inspection effort level and retail price, the manufacturer’s and retailer’s expected profits function, and the final customer’s consumer surplus.

Comment #4: 

In this article, we can not see how consumer utility and consumer behavior affect product market demand.

Response: Thank you very much for the valuable suggestions. In “3 The Model and Assumption”, we describe that the final customer’s quality utility is, and denotes the type of final customer, is the manufacturer’s product quality level; furthermore, denote the direct channel, the retail channel and the mixed channel respectively; then, we assume ～ uniform distribution, and the corresponding final customer’s consumer surplus is. 

Therefore, in direct channel, we can describe the final customer’s consumer surplus as = = (14)

In retail channel, we can describe the final customer’s consumer surplus as

 = = (30)

In mixed channel, we can describe the final customer’s consumer surplus as

 = + 

= (52)

Comment #5: 

In Mixed Channel, how does the product demand between the two channels affect each other?

Response: Thank you very much for the valuable suggestions. In our paper, we describe the mixed channel that the manufacturer sells their products in the traditional retail channel by the retailer, in the same time, the manufacturer sells their products to the final customer in direct channel (internet channel). The final customer’s product demand function will be ; denotes the market maximum demand, and is the product demand price elasticity. Therefore, In mixed channel, the manufacturer and the retailer’s stackelberg “leader-follower” control model can be described as follows: 

 = 

+ - - (31) 

s.t. 

 = (32)

The product demand in retail channel as 

The product demand in direct channel as 

 is the manufacturer’s product quality level in mixed channel, is the product demand price elasticity in retail channel, is the product demand price elasticity in direct channel, is the manufacturer’s wholesale price, is the retailer’s retail price, is the direct sale price in a direct channel, the manufacturer’s product quality cost function in direct channel is .

We have tried our best to improve the manuscript and made some substantial changes and necessary deletions according to the editors’ and reviewers’ comments. We earnestly appreciate the editors’ and reviewers’ professional work and hope that the corrections will make our manuscript suitable for publication in PLOS ONE. We are looking forward to receiving comments from reviewers in the future.

Once again, thank you very much for your valuable comments and suggestions.

Best wishes.

Yours sincerely,

Lilong Zhu (Correspondence)

E-mail: zhulilong2008@126.com

Tel: + 86 13853193366, Fax: + 86 531 8618 2769

School of Management, Shandong University, Ji’nan Shandong, 250100, China

College of Business, Shandong Normal University, Ji’nan 250014, Shandong, China

---

## [Decision Letter · Decision Letter 1]

16 Mar 2020

PONE-D-19-20228R1

Supply Chain Product Quality Control Strategy in Three Types of Distribution Channels

PLOS ONE

Dear Authors,

Thank you for submitting your manuscript to PLOS ONE. After careful consideration, we feel that it has merit but does not fully meet PLOS ONE’s publication criteria as it currently stands. Therefore, we invite you to submit a revised version of the manuscript that addresses the points raised during the review process.

Please see below

We would appreciate receiving your revised manuscript by 29.3.2020. To enhance the reproducibility of your results, we recommend that if applicable you deposit your laboratory protocols in protocols.io, where a protocol can be assigned its own identifier (DOI) such that it can be cited independently in the future. For instructions see: http://journals.plos.org/plosone/s/submission-guidelines#loc-laboratory-protocols

We look forward to receiving your revised manuscript.

Kind regards,

Dejan Dragan, PhD

Academic Editor

PLOS ONE

Reviewers' comments:

Reviewer's Responses to Questions

**Comments to the Author**

1. If the authors have adequately addressed your comments raised in a previous round of review and you feel that this manuscript is now acceptable for publication, you may indicate that here to bypass the “Comments to the Author” section, enter your conflict of interest statement in the “Confidential to Editor” section, and submit your "Accept" recommendation.

Reviewer #2: (No Response)

Reviewer #3: All comments have been addressed

2. Is the manuscript technically sound, and do the data support the conclusions?

Reviewer #2: Yes

Reviewer #3: Yes

3. Has the statistical analysis been performed appropriately and rigorously? 

Reviewer #2: Yes

Reviewer #3: Yes

4. Have the authors made all data underlying the findings in their manuscript fully available?

Reviewer #2: Yes

Reviewer #3: Yes

5. Is the manuscript presented in an intelligible fashion and written in standard English?

Reviewer #2: Yes

Reviewer #3: Yes

6. Review Comments to the Author

Reviewer #2: Really, the paper has been greatly improved as all comments provided by all other reviewers in the first round.

Reviewer #3: 1. It is a very nice thing that the new literature review part is added. But it seems to be to be too detailed. It will be better that the author just mentioned 2~3 related papers and on top of that, provide their own contributions. Moreover, those contributions should be more precise, instead of "..by providing a new

perspective.", the author need to be more specific about what exactly the new perspective is.

2. The model and assumption need to be more concise and concrete. For example, when the author mentioned the customer type follow a U[a,b], it needs more clarify about the value means and the types they are referring to. It will be better if the author check the model and assumption part one more time and provide additional explanation if necessary

7. PLOS authors have the option to publish the peer review history of their article (what does this mean?). If published, this will include your full peer review and any attached files.

Reviewer #2: Yes: Ali Wagdy Mohamed

Reviewer #3: No

---

## [Author Response · Author response to Decision Letter 1]

25 Mar 2020

Responses to Editors and Reviewers

Dear Professor Dejan Dragan and Reviewers,

Thank you very much for your suggestions and valuable comments about our paper submitted to PLOS ONE (Manuscript ID: PONE-D-19-20228R1). The manuscript’s title is “Supply Chain Product Quality Control Strategy in Three Types of Distribution Channels”.

We are also thankful to the reviewers for their critical reading and valuable comments on the manuscript. Those comments were very helpful for providing direction for our further studies. We have tried our best to revise our manuscript according to the comments. Attached, please find the revised version, which we would like to resubmit for your kind consideration. The main revised parts are marked in red in the paper. The following is a detailed explanation how we have complied with the editor’s and reviewers’ suggestions.

Responds to the editor’s comments:

Comment:

• A rebuttal letter that responds to each point raised by the academic editor and reviewer(s). This letter should be uploaded as separate file and labeled 'Response to Reviewers'.

• A marked-up copy of your manuscript that highlights changes made to the original version. This file should be uploaded as separate file and labeled 'Revised Manuscript with Track Changes'.

• An unmarked version of your revised paper without tracked changes. This file should be uploaded as separate file and labeled 'Manuscript'.

Response: Thank you very much for the highly praises and the valuable suggestions for this manuscript. According to the editor’s and reviewers’ suggestions, we have completed all the revision which each point raised by the academic editor and reviewers, the file is labeled 'Response to Reviewers'. We have completed a marked-up copy of our manuscript that highlights changes made to the original version, the file is labeled 'Revised Manuscript with Track Changes'. We also have completed an unmarked version of our revised paper without tracked changes, the file is labeled 'Manuscript'.

Comment #2: 

Response: Yes, I would like to make the peer review history publicly available.

Responds to the reviewer’s comments:

Comment #1:

If the authors have adequately addressed your comments raised in a previous round of review and you feel that this manuscript is now acceptable for publication, you may indicate that here to bypass the “Comments to the Author” section, enter your conflict of interest statement in the “Confidential to Editor” section, and submit your "Accept" recommendation.

Reviewer #2: (No Response)

Reviewer #3: All comments have been addressed

Response: Thank you very much for the valuable suggestions. We have addressed and completed all the revised comments in the new 'Manuscript'.

Comment #2:

Is the manuscript technically sound, and do the data support the conclusions?

Reviewer #2: Yes

Reviewer #3: Yes

Response: Thank you very much for the valuable suggestions. In this manuscript, all the data support the conclusions.

Comment #3:

Has the statistical analysis been performed appropriately and rigorously?

Reviewer #2: Yes

Reviewer #3: Yes

Response: Thank you very much for the valuable suggestions.

Comment #4:

Have the authors made all data underlying the findings in their manuscript fully available?

Reviewer #2: Yes

Reviewer #3: Yes

Response: Thank you very much for the valuable suggestions.

Comment #5:

Is the manuscript presented in an intelligible fashion and written in standard English?

Reviewer #2: Yes

Reviewer #3: Yes

Response: Thank you very much for the valuable suggestions. We made necessary revisions and language editing in the manuscript according to your suggestions, and the English has also been edited by Wiley English Language Editing Services.

Comment #6:

Review Comments to the Author.

Reviewer #2: Really, the paper has been greatly improved as all comments provided by all other reviewers in the first round.

Reviewer #3: 1. It is a very nice thing that the new literature review part is added. But it seems to be to be too detailed. It will be better that the author just mentioned 2~3 related papers and on top of that, provide their own contributions. Moreover, those contributions should be more precise, instead of ". by providing a new perspective, the author need to be more specific about what exactly the new perspective is.

2. The model and assumption need to be more concise and concrete. For example, when the author mentioned the customer type follow a U[a,b], it needs more clarify about the value means and the types they are referring to. It will be better if the author check the model and assumption part one more time and provide additional explanation if necessary.

Response to Reviewer #2: Thank you very much for the valuable suggestions. We have revised and completed all comments provided by the editor and reviewers in the first round.

Response to Reviewer #3: Thank you very much for the valuable suggestions.

1. We have revised the literature review part, which is chiefly related to three streams of literature, it seems not to be too detailed, which is mentioned the related papers and on top of that, provide their own contributions. Moreover, those contributions should be more precise as line 150-160.

In this paper, first of all, we will introduce the distribution channel demand elasticity ratio and investigating how to construct a product quality control strategy model and channel coordination in three types of distribution channels (direct channel, retail channel, and mixed channel); what’s more, we consider the manufacturer’s product quality decision and quality prevention strategy, the retailer’s product pricing decision and quality inspection strategy, and the final customer’s product demand decision in a three-echelon supply chain; above all, we also establish a product quality control strategy model in three types of distribution channels to eliminate the influence of “channel conflict” and “free-riding behavior”, which will improve the manufacturer’s and retailer’s expected profits and the final customer’s consumer surplus.

2. We have revised model and assumption more concise and concrete in line 174-l96. 

For example, The final customer’s quality utility is , and denotes the type of final customer; then, we assume Ɵ~U[a, b] uniform distribution, i.e. a is the final customer lower limit of distribution quantity, b is the final customer upper limit of distribution quantity, and the corresponding final customer’s consumer surplus is .

We have checked the model and assumption part one more time, and then, provided additional explanation more clearly and accurately.

We have tried our best to improve the manuscript and made some substantial changes and necessary deletions according to the editors’ and reviewers’ comments. We earnestly appreciate the editors’ and reviewers’ professional work and hope that the corrections will make our manuscript suitable for publication in PLOS ONE. We are looking forward to receiving comments from reviewers in the future.

Once again, thank you very much for your valuable comments and suggestions.

Best wishes.

Sincerely yours,

Lilong Zhu

---

## [Editor Report · Decision Letter 2]

31 Mar 2020

Supply Chain Product Quality Control Strategy in Three Types of Distribution Channels

PONE-D-19-20228R2

Dear authors,

We are pleased to inform you that your manuscript has been judged scientifically suitable for publication and will be formally accepted for publication once it complies with all outstanding technical requirements.

With kind regards,

Dejan Dragan, PhD

Academic Editor

PLOS ONE

Additional Editor Comments (optional):

The improvements have been noticeable so that the paper deserves to be published in the Plos One.
---

## [Editor Report · Acceptance letter]

6 Apr 2020

PONE-D-19-20228R2 

Supply Chain Product Quality Control Strategy in Three Types of Distribution Channels 

Dear Dr. Zhu:

I am pleased to inform you that your manuscript has been deemed suitable for publication in PLOS ONE. Congratulations! Your manuscript is now with our production department. 

With kind regards,

on behalf of

Dr. Dejan Dragan 

Academic Editor

PLOS ONE